# Development of a family-centered intervention to support self-determination in adolescents and young adults with intellectual disability in home environments: Protocol for a multistage mixed methods design

Sergi Fàbregues[1,2], Araceli Arellano[3], Ahtisham Younas[4*], Eva Vicente[5],
Ana Berástegui[6], Ana Casas[7], Elsa Lucia Escalante-Barrios[8], Clara Andrés-Garriz[2],
Cristina Mumbardó-Adam[2]

1 Department of Psychology and Education, Universitat Oberta de Catalunya, Barcelona, Spain,
2 Department of Cognition, Development and Educational Psychology, Faculty of Psychology, Universitat de Barcelona, Barcelona, Spain, 3 Department of Education, Faculty of Education and Psychology, Universidad de Navarra, Pamplona, Spain, 4 Faculty of Nursing, Memorial University of Newfoundland, St. John's, NL, Canada, 5 Department of Psychology and Sociology, Faculty of Education, Universidad de Zaragoza, Zaragoza, Spain, 6 University Institute of Family Studies, Universidad Pontificia de Comillas, Madrid, Spain, 7 Faculty of Education and Sport, Universidad de Deusto, Bilbao, Spain, 8 Department of Education, Universidad del Norte, Barranquilla, Colombia

* ay6133@mun.ca

## Introduction

In recent years, the concept of self-determination has gained currency in the field of intellectual and developmental disabilities, particularly with regard to individuals with intellectual disability. According to the American Association on Intellectual and Developmental Disabilities, intellectual disability is "a condition characterized by significant limitations in both intellectual functioning and adaptive behavior that originates before the age of 22" [1]. Alternatively, the American Psychiatric Association's DSM-5 manual [2] defines intellectual disability as involving "impairments of general mental abilities that impact adaptive functioning in three domains, or areas," namely, conceptual, social, and practical domains. Self-determination, in the context of disability, is defined by Shogren et al. [3] as "a dispositional characteristic manifested as acting as the causal agent in one's life".

Extensive research has been conducted on self-determination among people with intellectual disability, addressing, among other topics, the perception and experience of self-determination by these people and their family members [4]; the conceptualization and operationalization of self-determination by practitioners and other stakeholders [5]; the measurement of self-determination across cultures [6,7]; the facilitators of and barriers to self-determination in different contexts [8,9]; and the development and evaluation of interventions to facilitate and enhance self-determination [10,11].

Increased self-determination has been shown to enhance the quality of life of individuals with intellectual disability. Numerous studies have shown a positive relationship between various key indicators of the quality of life of individuals with intellectual disability and self-determination. These indicators include academic achievement [12], school and

**Data availability statement:** Since this manuscript reports a study protocol, and not a completed study, no datasets have yet been generated and analyzed in the context of this study. All relevant data from this study will be made available upon study completion. Specifically, transcriptions of the interviews and focus groups, as well as the data generated in the concept mapping, will be made publicly available via the Internet in the Open Science Framework (OSF) data repository. This information will not include the participants' names or other personally identifiable information, ensuring that the shared data will be free of any information that could link the participants' responses to them. The informed consent will include a statement explaining to participants that an anonymized copy of their data will be made available online and that these data may be used for secondary analysis. In this consent form, each participant will be given the opportunity to opt out of both of these options. Therefore, participants will not be re-contacted to seek permission to share or reuse their data. A data sharing plan has been developed by the research team.

**Funding:** This study is funded by a Research Accelerator Grant 2024 awarded in February 2025 by the Universitat Oberta de Catalunya (PI: Sergi Fàbregues). The funder had no role in study design, data collection and analysis, decision to publish, or preparation of the manuscript.

**Competing interests:** The authors have declared that no competing interests exist. Funding This study is funded by a Research Accelerator Grant 2024 awarded by the Universitat Oberta de Catalunya (PI: Sergi Fàbregues). The funder had no role in study design, data collection and analysis, decision to publish, or preparation of the manuscript.

post-school outcomes [13], psychological well-being [14], and employment [15]. This construct is operationalized through a variety of skills, such as decision-making, risk-taking, self-advocacy, and self-management strategies. The likelihood of people with disabilities implementing these strategies and improving their quality of life will depend on the availability of adequate systems of individualized supports, defined as "resources and strategies that aim to promote the development, education, interests, and personal well-being of a person and that enhance individual functioning" [16].

From a person-environment perspective of disability, the development of support systems to promote self-determination for individuals with intellectual disability is closely related to the extent to which environmental factors, such as personal, interpersonal, and societal contexts, contribute to the establishment of such systems [17]. Among these contexts, the role of the family is decisive in facilitating or impeding self-determination in these individuals [15,18–26]. Parents are responsible for teaching children in their early years how to behave and interact with others in areas such as developing and expressing self-determined beliefs; knowledge; and skills such as problem-solving, self-advocacy, and goal setting [27]. Since parents serve as role models, children's willingness to act in a self-determined manner is directly affected by how self-determination is expressed, developed, and reinforced in the family [23]. Furthermore, parents are responsible for laying the foundations for their children's future self-determination by developing learning opportunities and creating a supportive home environment.

## Study rationale

Although researchers have generally acknowledged the importance of the role families play in supporting self-determination in children with intellectual disability, more research is needed, particularly on the ways in which families can facilitate this support in the home environment. Following Bradley [28] and Bradley et al. [29], in this study, we conceptualize the home environment as encompassing both the physical setting of the home and social-psychological factors. These factors include the interactions between parents and their sons and daughters, the dynamics among household members, family routines, the degree of family cohesion, and the performance of regulatory activities by parents, among others. From this perspective, home-based involvement activities extend beyond the physical environment of the home itself and may include activities with household members, such as going to the movies or museums, participating in outdoor activities, or shopping for food or clothing [30].

A recent scoping review by Dean et al. [31] identified 24 studies that examined the ways in which families of youth with intellectual and developmental disabilities supported the development of self-determination. While most of the studies reviewed focused on the family's view of the importance of self-determination to their children and to adolescents with disabilities of this sort, only a few of these provided evidence on how families promoted self-determination in the home context. This gap in research merits further attention since most parenting activities take place in the home environment, where naturally occurring and induced contextual changes can potentiate their children's opportunities for making decisions, becoming more independent, and enhancing their self-esteem [32]. Brotherson et al. [19] conducted a

qualitative study involving semi-structured interviews and observations with families of young children with disabilities to examine how the home environment influenced the strategies they used to support their children's self-determination. The authors identified several strategies used by these families, such as encouraging children to choose what they want to eat, creating spaces for play and interaction with others, supporting independent movement around the house, teaching goal achievement during play with toys, and providing opportunities for children to control their personal space and privacy [19,22]. Another notable gap identified in Dean et al.'s [31] review was the scarcity of interventions focused on family support and self-determination, with only three studies reporting such interventions. The scarcity of interventions, specifically in Spain, was also highlighted in a previous study by Arellano and Peralta [18] that examined Spanish parents' attitudes towards young children with intellectual disability.

## Study purpose and research questions

This paper reports the protocol for a multistage mixed methods study. In order to address the gaps in the body of research discussed above, the purpose of this mixed methods study is to determine how self-determination develops and is supported in home environments in adolescents and young adults with intellectual disability and a mild or moderate level of support, with the aim of developing an intervention to promote self-determination in these environments. The study is based on a family-centered approach, thus considering the family as a unit of intervention and as the key factor in implementing the study. This approach relies on collaboration between families and professionals to achieve the design of the intervention. We aim to answer the following research questions:

Research question 1: How do families of adolescents and young adults with intellectual disability support the self-determination of these individuals in their home environments in Spain?

Research question 2: What context-sensitive strategies and practical actions are identified by professionals, adolescents and young adults with intellectual disability, and their families as relevant and feasible to support the self-determination of adolescents and young adults with this disability in home environments in Spain?

Research question 3: What are the main elements of a desirable context-sensitive intervention to support the self-determination of adolescents and young adults with intellectual disability in their home environments in Spain?

Research question 4: What strategies should be used to implement this intervention?

## Theoretical framework

This study is based on the Causal Agency Theory, developed by Shogren et al. [3] to explain how people (with and without disabilities) become self-determined in the course of their lives. Drawing on the principles of strengths-based approaches to disability, positive psychology, and the social-ecological model of disability, which emphasizes the interaction between individuals and their environment, this theory constitutes a valuable framework for explaining how individuals, at different stages of their life, "define the actions and beliefs necessary to engage in self-caused, autonomous action that addresses basic psychological needs" [33]. According to this theory, self-determined individuals are causal agents who "act in service to freely chosen goals that propel action" [17]. This theory has three main elements: volitional actions, agentic actions, and action-control beliefs (i.e., to decide, to act and to believe). Specific skills associated with volitional action include choice-making, decision-making, goal setting, problem-solving, and planning; those associated with agentic action include self-management, goal-attainment, problem-solving, and self-advocacy; and those related to action-control beliefs include self-awareness and self-knowledge.

## Methods and analysis

### Study design

Using a multistage mixed methods design [34,35], this study will include three sequential phases: Phase 1, a grounded theory study involving the families of individuals with intellectual disability, and addressing research question 1; Phase 2,

a participatory group concept mapping involving adolescents and young adults with intellectual disability, their families, and professionals, and addressing research question 2; and Phase 3, based on intervention mapping, and addressing research questions 3 and 4. Integration of the three phases will occur through *connecting* (inviting participants in each phase to participate in the subsequent phases); through *building*, using the findings from Phase 1 to generate the statements used in Phase 2, and through *merging*, by synthesizing the findings from Phases 1 and 2 to develop the intervention in Phase 3 [36].

Similar approaches to the iterative development of complex interventions in the field of intellectual and developmental disabilities have been used recently, and have shown clear advantages, such as the collaboration between families and professionals in the development of the intervention and the opportunity to maximize the feasibility, utility, and potential effectiveness of the intervention as a result of this collaboration [37]. O'Cathain et al.'s [38] guidelines for actions to consider when developing complex interventions will be considered in the development of the intervention built in this study. These guidelines, recently incorporated into the Medical Research Council's (MRC) Framework [39], provide a clear protocol for incorporating theory and evidence into interventions in line with the rationale of this study. Fig 1 shows a summary of the study design and the actions suggested by O'Cathain et al. [38] for conceptualizing and planning complex interventions. The study has been registered in the Open Science Framework (OSF) (https://doi.org/10.17605/OSF.IO/M94DX).

## Phase 1: Grounded theory

**Design.** We will use a grounded theory approach to address research question 1. Grounded theory is a qualitative research method that generates a substantive theory explaining a set of actions related to a given phenomenon that is carried out by a particular group of individuals [40]. By using this approach, we aim to build a theoretical model that describes the process by which families support the self-determination of their adolescents and young adults with intellectual disability within their home environments.

Of the various versions of grounded theory proposed in the literature, we will use Strauss and Corbin [41] for three reasons. First, this version adheres to well-defined procedures that enable researchers to "build a theory that derives organically yet as rigorously and systematically as possible from the setting under investigation" [42]. Through semi-structured interviews and a photo elicitation exercise, we aim to extend our understanding of the process of developing self-determination in the home environment by exploring the barriers and facilitators unique to this environment, as well as the strategies parents use to support self-determination. Second, this grounded theory version is particularly well suited to examining the actions and interactions of individuals, while also considering the immediate and broader contexts in which these actions occur [43]. Developing and enhancing self-determination in the home is the outcome of a series of interactions within the family that either facilitate or inhibit the creation of a supportive environment for the person with intellectual disability. At the same time, these interactions are shaped by context-specific factors that can be either internal or external to the family and home environment. Third, the Strauss and Corbin version of grounded theory supports the use of an early literature review to inform the study's rationale and research questions and to cultivate theoretical sensitivity throughout the research process without hindering the emergence of substantive theory [44–46]. In the early stages of protocol development, the first (SF), second (AA), and last authors (CMA) conducted a non-systematic review of the literature on the role of the family in the self-determination of people with disabilities. In addition, AA and CMA are internationally recognized experts in the field of self-determination of people with intellectual disability and therefore have extensive knowledge of the literature on this topic.

In line with the third reason stated above, we will employ Thornberg's informed grounded theory approach [47], which emphasizes the role of previous theories in the development of the grounded theory yet allows for the emergence of new inductive categories. Accordingly, Causal Agency Theory will be integrated at multiple stages of Phase I: during data collection to inform the questions included in the interview guide, and during data analysis by employing its concepts as

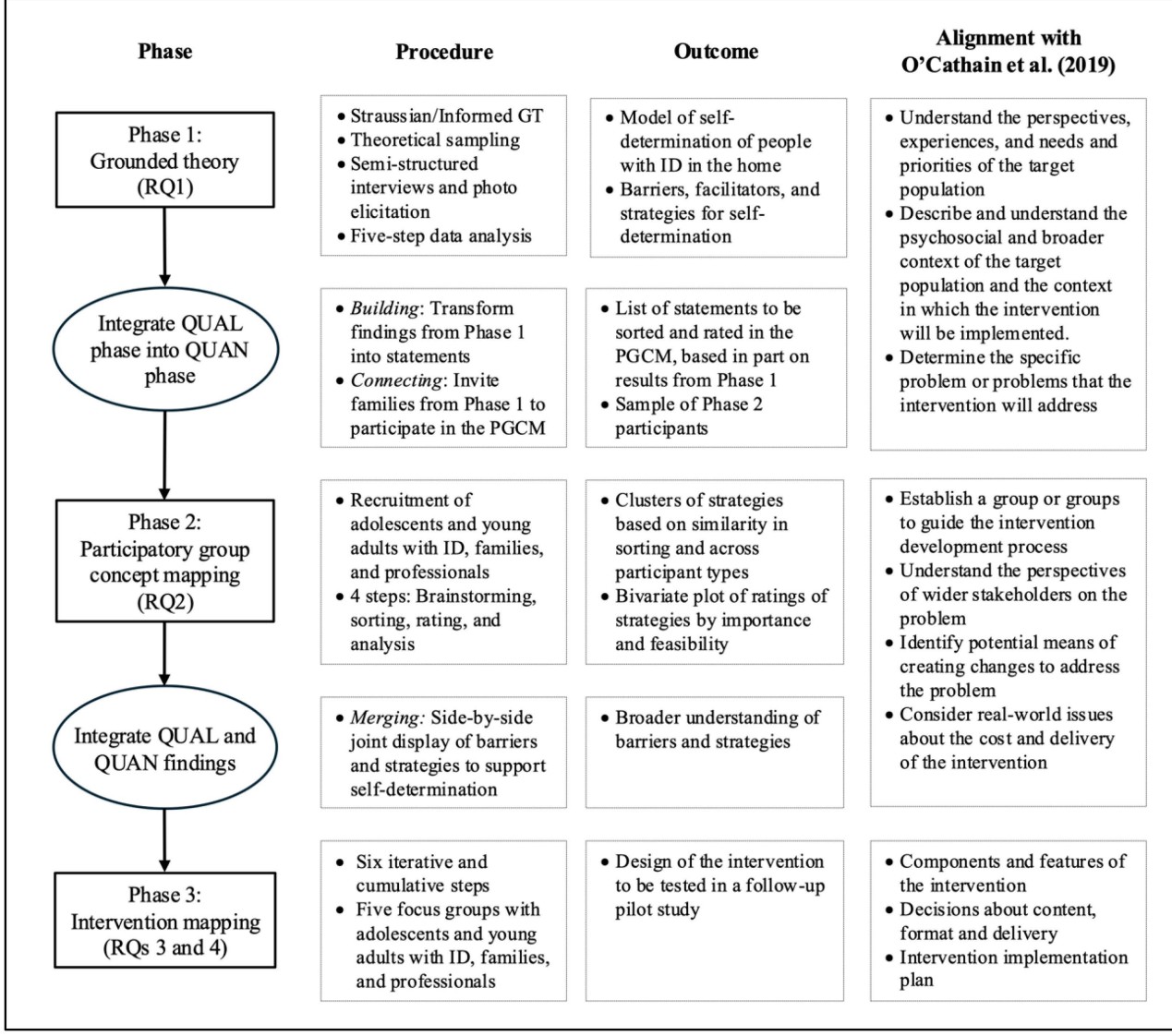

**Fig 1. Multistage mixed methods design.** Note. RQ = Research Question, QUAL = Qualitative, QUAN = Quantitative, GT = Grounded Theory, PGCM = Participatory Group Concept Mapping, ID = Intellectual Disability.

"sensitizing concepts" to guide comparisons and aid in developing the substantive theory. As a result of this process, our goal is to refine and extend Causal Agency Theory by applying it to the family and home environments.

**Sampling and recruitment.** Theoretical sampling will be used to select a sample of participating families. The study will include families residing in Spain with 16–22-year-old adolescents and/or young adults with intellectual disability who have mild or moderate support needs and are receiving special education services. In each family, interview data will be collected from one or both parents. If the primary caregiver of the person with intellectual disability is someone other than the parents (e.g., the grandmother), that person will be interviewed either with the parents or alone. For the families to be included in the study, adolescents or young adults with intellectual disability must (1) be formally diagnosed with an intellectual disability, either according to their IQ level (for older diagnoses) or according to their adaptive behavior

and IQ level for diagnoses based on the DSM-5; (2) require a moderate or low level of support, based on the Spanish versions of the Supports Intensity Scale (SIS) [48] or the Supports Intensity Scale for Children and Adolescents (SIS-C) [49]; and (3) if they have another type of disability, it must be a mild sensory impairment, such as partial blindness, or partial deafness. Autism spectrum disorder, sensory processing disorders, and other types of developmental disabilities will be excluded. The theoretical sampling strategy will be combined with the strategy of stratified purposive sampling [50], which consists of "a hybrid approach in which the aim is to select groups that display variation on a particular phenomena but each of which is fairly homogeneous" [51]. The latter strategy is intended to facilitate the representation of different socio-economic groups. While this strategy, due to its qualitative nature, will not allow for full generalization or representativeness of results, it will allow for the identification of variations across socioeconomic strata. Such findings may help to broaden the scope of the intervention to be developed in Phase 3 of the study, particularly in terms of the population to which the intervention is applicable.

The participants will be recruited in three stages. First, a series of institutions and special education schools with which the co-authors frequently collaborate, located in the Autonomous Communities of Catalonia, the Basque Country, Cantabria, Aragon, Madrid and Navarra, will be contacted. To ensure diversity among the recruited families, these institutions and special education schools will be selected from various socioeconomic areas using socioeconomic indices for small areas in each Autonomous Community when available. Variables incorporated in these indices typically encompass employment status, educational attainment, immigration status, and income of all individuals residing in each of the territorial units. If the institutions and special education schools agree to collaborate in the study, one person from each institution will be asked to act as the primary interlocutor between the potential participants and the researchers. Second, a recruitment flyer will be created that includes the following elements: a brief title of the study, the purpose of the study, the eligibility criteria, what participation in the study will entail, the potentially positive outcomes of the study findings, and how to contact the researchers. In addition, the flyer will state that participants will receive a $20 Amazon gift card to participate in this phase of the study. The flyer will be written in plain language and will include illustrations. The contact person at each institution will be asked to distribute the flyer to individuals and families eligible to participate in the study. Third, upon receipt of an expression of interest from a family or individual, the researchers will contact them by phone or email to confirm their eligibility according to the participant inclusion criteria described above. A screening eligibility form will be used that includes the type of disability diagnosis, the nature and severity of the disability, and the type of support needed (see above for instruments used). If the potential participants meet the inclusion criteria, we will provide them with a participant information sheet that includes an informed consent form to be signed. The information sheet will contain details about the study, including the purpose, the subsequent steps for initiating data collection with that particular family, and information on how ethical issues will be addressed (i.e., procedures for maintaining confidentiality and anonymity, how data will be handled, and the ability to withdraw from the study at any time).

In accordance with the principles of grounded theory and theoretical sampling, recruitment of participants will continue until data saturation has been achieved. Saturation is a specific concept of grounded theory that implies that the researchers shall continue to recruit new participants until no new themes or concepts have been identified in the analysis of data [52]. Thus, once an initial group of families has been interviewed and their information has been analyzed, additional families will be recruited until the newly interviewed families do not generate new themes, that is, those not identified in the previous rounds of analysis. Although it is difficult to estimate the number of participating families owing to the particularities of saturation, in line with what has been suggested in the literature, we aim to recruit between 15 and 20 families [40].

**Data collection.** Data will be collected using semi-structured interviews and photo elicitation, either in the home of the participating families or online (when the first option is not possible). Due to their relatively and emergent nature, semi-structured interviews are ideal for gathering detailed information about participants' experiences and emotions while allowing them maximum freedom and guiding them towards the constructs of interest [53]. Photo elicitation is a qualitative interview technique in which photographs are used to elicit dialogue and discussion during the interview

[54]. The photo elicitation method has been recognized for its potential to yield a wealth of information by more actively engaging participants, generating naturalistic insights into the phenomenon under study, and facilitating rapport-building [55,56]. In this study, photo elicitation will be particularly useful in obtaining in-depth experiential knowledge about perceptions, emotions and everyday practices regarding self-determination that are not discussed in conventional semi-structured interviews [57]. This approach will facilitate a deeper understanding of the family dynamics of self-determination within each household and thus provide a more contextualized view of phenomena that would otherwise be too difficult to capture. In addition, this approach should allow us to obtain a more authentic representation of self-determination practices in the home environment, as visual documentation will be generated from the participants' perspectives.

In each family, two interviews, including one photo elicitation interview, will be conducted with one or both parents. While the number of participants from each family may vary between interviews (e.g., the first interview may include both the mother and father, while the second interview may include only the mother or only the father), at least one participant (i.e., the same person) must participate in the two interviews. During the first interview, we will focus on introducing the research project and ethical principles, as well as gathering descriptive and exploratory information about socio-demographic data and family structure, the adolescent's or young adult's disability history, and the parents' perspectives and experiences with their son's or daughter's daily activities and practices, among other topics. Additionally, we will focus on how they support their son or daughter self-determination, with some of the questions guided by the concepts of the Causal Agency Theory. The interview will include an open-ended question asking participants to define self-determination, followed by more specific questions focusing on aspects such as the parents' views on the level of autonomy of the adolescent or young adult in everyday life; how he or she is involved in decision-making processes; his or her propensity to initiate activities, try new things or set goals; whether parents support their sons and daughters in pursuing their goals; and the barriers encountered by families, among others. Despite this organization of the questions, to be consistent with the emergent nature of grounded theory, what we will ask in the interviews and how we will ask it will likely vary slightly between and within interviews. Prompts and probes will be used to elicit additional information from participants.

After the first interview, each family will be given a photo elicitation task. Families will be given the option of using their cell phone camera, another camera of their choice, or a disposable camera. Each family will be provided with a verbal explanation of the photo elicitation procedure, a written protocol of the steps, and the following instructions: *Take pictures that reflect the development of their ability to decide, to act and to believe in themselves*. Ethical guidelines for obtaining permission from non-household members who may appear in the photographs will also be provided. Photographs will be requested from families 10–20 days after the interview, and two reminders will be sent during this period. If the photos were taken with a cellphone, participants will be given the option to send them via email or Microsoft OneDrive. Since absolute confidentiality cannot be guaranteed over the Internet, the disposable camera option will allow photos to be processed only on paper. In the second interview, participants will be requested to choose from four to six photos to be included in a PowerPoint slideshow. For each photo, participants will be asked to label and answer questions about who took it, when it was taken, what elements may be present in or absent in the photo, the reasons for taking the photo, their feelings when looking at it, and its symbolic meaning and characteristics.

Interviews will last between one hour and 90 minutes and will be conducted by the members of the research team. Only one researcher will be present during each interview, and the participants will be asked for permission to record the interview before it begins. The interviews will be recorded using a recording application on the researchers' cellphones. A pilot study with three families will be conducted at the beginning of this phase to test the data collection strategy and make changes if necessary.

**Data analysis.** The analysis will be conducted concurrently with data collection, according to the principles of theoretical sampling and data saturation discussed above. Thus, as the analysis progresses, the resulting findings will inform the selection of new participants to address any lacunae observed in the course of the study in terms of several key characteristics of the participants, or to clarify or provide further insights into the theoretical concepts identified during

the analysis. The audio files, photographs, and interview transcriptions will be imported into MAXQDA 2022, which will be used to analyze the data.

Data analysis procedures described by Strauss and Corbin [41] will be used in a five-step procedure: (1) open coding, (2) axial coding, (3) selective coding, (4) integration of an existing theoretical framework into the theory that has been generated so far, and (5) final generation of a substantive theory. We will move through these stages iteratively, constantly making use of using comparative analysis, a process that involves generating increasingly more abstract ideas by comparing different instances of data and identifying similarities and differences. As we have previously described, in the course of constant comparative analysis, the concepts borrowed from Causal Agency Theory will be used as "sensitizing concepts", therefore allowing this framework to establish "a theoretical dialogue with the data" [58]. SF, AA, and CMA will be involved in the data analysis.

In Step 1, open coding will involve line-by-line coding of the transcripts, by breaking the data down into words and phrases, while assigning descriptive labels. In labeling the codes, we will use gerunds to reflect the actions described by the participants in their home environments to support self-determination, focusing on the process rather than the description. After generating open codes, we will engage in an iterative process of integrating these codes into concepts, synthesizing these concepts into categories and subcategories, and assigning labels that reflect the properties and dimensions of each category.

In Step 2, axial coding will involve conceptualizing, in an iterative and abductive manner, the action patterns involved in the process of supporting self-determination described by the participants using the coding paradigm strategy outlined by Strauss and Corbin [41]. Specifically, this phase of coding will consist of identifying a central phenomenon in the data and determining the relationships among the categories and subcategories identified during the open coding phase in terms of the conditions, contexts, strategies (participants' actions/interactions), and consequences associated with this central phenomenon.

In Step 3, selective coding will involve selecting a core category and linking that category to other categories that would have been created earlier, with the aim of generating a first draft of the emerging grounded theory. In Step 4, Causal Agency Theory will be integrated with the emerging theory in order to compare them and assess their fit, contextualize and add explanatory power to the emerging theory, and refine Causal Agency Theory to address self-determination in the specific context of the home environment [59]. As a result, in Step 5 we will generate the final emerging theory explaining the process of supporting self-determination in adolescents and young adults with intellectual disability within the home environment, including barriers and facilitators to self-determination and the strategies used by families for supporting it.

## Phase 2: Participatory group concept mapping

**Design.** Participatory group concept mapping methodology [60,61] will be used to answer research question 2 by identifying context-sensitive strategies and practical actions that support self-determination among adolescents and young adults with intellectual disability in the home environment in Spain. This methodology provides systematic procedures for gathering, organizing, and ranking stakeholders' opinions on a specific issue. Stakeholder views are graphically represented in several two-dimensional maps. Recent studies have shown that participatory group concept mapping findings can serve as a foundation for developing interventions [62,63]. In addition, unlike similar approaches such as the Delphi or Nominal Group techniques, which focus on attaining consensus on a particular issue, participatory group concept mapping is particularly useful for comparing the perspectives of different groups on the phenomena being examined. This feature is particularly important to this study, as we anticipate identifying diverse understandings of self-determination among adolescents and young people with intellectual disability, their families, and professionals. This phase will be completed entirely online. Participants will receive a $10 Amazon gift card to participate in this study phase.

**Sampling.** Community partners, including adolescents and young adults with intellectual disability, their parents or primary caregivers, and professionals working with them in Spain, will be recruited. The inclusion criteria for the

parents and primary caregivers will be the same as those in Phase 1. In the specific case of the adolescents and young adults with intellectual disability, they will need to have sufficient communication and comprehension skills to be able to participate. Information about their skills will be gathered from their parents or caregivers. Professionals will need to have at least two years of experience in (1) supporting the development of self-determination in adolescents and/or young adults (16–22 years old) with intellectual disability or (2) working with families of adolescents and/or young adults (16–22 years old) with intellectual disability.

We intend to start implementing the participatory group concept mapping with a sample of approximately 100 participants in the first step (brainstorming – see the four steps below). Although we will strive to maintain the participant count throughout the subsequent two steps (sorting and rating), the number of participants will probably decrease after the first step. Such a pattern of sample attrition was found in Donnelly's [64] review of concept mapping dissertations, which reported averages of 48, 27, and 34 participants in the brainstorming, sorting, and rating steps, respectively. In light of potential attrition, we will aim to have at least 30–35 participants in steps 2 and 3, and we will recruit new participants if necessary. These numbers are consistent with recommendations in the literature on participatory group concept mapping methodology [60,64].

Participants will be identified and recruited using three complementary strategies. First, we will contact institutions and special education schools in a way that is similar to what was done in Phase 1, including the goal of promoting socioeconomic diversity. Second, families who took part in the grounded theory phase will be invited to participate in this second phase. Third, using a snowball sampling strategy, these families will be asked to provide the names of other families with similar characteristics and the names of professionals who work with them. Once potential participants have been identified, the same screening eligibility form used in Phase 1 will be used to confirm the eligibility of adolescents and young adults with intellectual disability and their parents or primary caregivers. In the case of professionals, a new screening eligibility form will be developed that includes items related to the length and nature of their professional experience. Communication with potential participants at this point will be by phone or email.

**Data collection and analysis.** As briefly mentioned above, data collection and analysis will be carried out in four steps: (1) brainstorming, (2) sorting, (3) rating, and (4) analysis. Internet-based groupwisdom™ concept mapping software will be used in all these steps.

In Step 1, the participants will be contacted via email and receive a unique username and password to access the software. After reading the online informed consent form, they will either accept or decline participation. For adolescents and young adults with intellectual disability, a trusted person who is well known to the participant will explain the content of the informed consent form in a helpful way. This trusted individual will also accompany the person with intellectual disability through the various steps that are part of this phase. All documents distributed to this type of participants during this phase will be in an easy-to-read format to facilitate comprehension. Participants will then be directed to a secure website, where they will answer a series of demographic questions before being asked the following focus question: *To ensure that adolescents and young adults with intellectual disability have support for self-determination in the home environment, the following strategies MUST be used...* In responding to this focus question, participants will answer using complete written sentences. Each participant will be allowed to generate as many statements as they wish and will be able to see the statements generated by the other participants. In this and in the next phase, emails will be sent to remind participants of the need to complete each phase within 30 days.

In Step 2, the statements generated during brainstorming will be integrated with two additional data sources. First, they will be integrated with the results of a recent Delphi study on ways to support families of people with intellectual disability to promote their self-determination, led by some of the co-authors of this study protocol [65]. Although this Delphi study focused on interventions at a general level and not specifically in the home environment, and although participation was limited to professionals, several of the components identified in the findings could be relevant to the purposes of our study. Second, the statements will also be integrated with the qualitative findings from Phase 1 about self-determination barriers

and facilitators and the strategies families use to support it. To this end, we will use a *building* integration strategy through a joint display instrument development table [66], which will allow us to systematically transform each qualitative finding of interest into specific statements to be used in Step 3. Once all the potential statements from the different sources have been completed, we will merge them, removing those that are redundant. Using the same secure website, participants will next be asked to sort the consolidated statements resulting from this integration into conceptually relevant piles and to identify each pile with a word or phrase that best reflects the concepts in that pile. Then, in Step 3, using a five-point Likert scale, participants will rate their agreement with each statement based on its importance (*How important are each of the following strategies or practices in...?*), and feasibility (*How achievable are each of the following strategies or practices in...?*).

In Step 4, the groupwisdom™ software will be used to analyze the data. Several procedures will be required to complete the analysis. First, the sorted statements will be analyzed using non-metric multidimensional scaling. Multidimensional scaling will generate a two-dimensional map of the statements based on the criterion that statements that sort together more frequently are closer together in a two-dimensional space. Second, agglomerative hierarchical cluster analysis will be used to divide all strategies and actions into multiple clusters in a cluster map. The average rating data for each strategy and cluster will be calculated. This will provide a visual representation of the strategies between and within clusters, as well as the relative importance of each cluster compared with the others. Third, after creating the cluster maps, we will conduct two distinct analyses: (1) we will compare the level of agreement between the cluster ratings of adolescents and young adults with intellectual disability, their families, and practitioners using the Pearson correlation coefficient, and (2) we will generate a bivariate plot of the strategies' ratings according to their importance and feasibility, separated into quadrants above and below each rating variable's mean. The upper-right quadrant of the plot will represent the strategies that are above average in both importance and feasibility within a cluster, thereby enabling us to determine which strategies within each cluster should be prioritized to support self-determination.

## Phase 3: Intervention mapping

**Design.** To answer research questions 3 and 4, in Phase 3 we will use an adapted intervention mapping approach [67] to develop a context-sensitive intervention for supporting self-determination within the home environment. This approach consists of six iterative and cumulative steps aimed at guiding the planning, development, and implementation of programs and interventions according to the needs of the target groups. This adapted version of intervention mapping will involve, in several of these steps, the participation of adolescents and young adults with intellectual disability, their families, and professionals. If possible, the same participants from Phases 1 and 2 will be recruited for Phase 3; alternatively, new participants with similar characteristics to those in the previous phases can be recruited. Recruitment procedures will be the same as in Phase 2.

**Data collection and analysis.** In Step 1, *needs assessment*, our aim will be to summarize the facilitators and barriers faced by families of adolescents and young adults with intellectual disability in supporting self-determination in their home environments, and we will also summarize the strategies families and professionals might use to achieve this support. To this end, the qualitative and quantitative findings from Phases 1 and 2, will be integrated through side-by-side joint displays [36,68]. This will involve bringing the two types of findings together in a table, and for each facilitator, barrier and strategy, generating an integrated interpretation or meta-inference. These meta-inferences will provide us with a more comprehensive understanding of the facilitators, barriers and strategies beyond the individual findings of Phases 1 and 2. At this point, the Theoretical Domains Framework (TDF) [69], a theory-informed framework for identifying affective, cognitive, environmental and social determinants of behavior, will be used to better understand and organize the individual-level factors that facilitate or hinder behavior change in families with respect to supporting the self-determination of their adolescents and young adults with intellectual disability.

After integrating the findings, three focus groups of 10 participants each will be held with adolescents and young adults with intellectual disability, along with their families and the professionals. These focus groups will be moderated by one of the three primary investigators (SF, AA, and CMA) and will focus on the presentation of results from Phases 1 and 2, along with the integrated findings, to the participants. Additionally, they will focus on facilitating reflection on the practical issues related to intervention activities and goals. The focus groups will be audio recorded and transcribed using the same procedures described in Phase 1. Thematic analysis according to Boyatzis [70] will be used by SF, AA, and CMA to analyze the focus group transcripts. The analysis will comprise four phases. In Phase 1, the researchers will familiarize themselves with the data by thoroughly reading the transcripts and preparing a summary of the major themes identified in each transcript. In Phase 2, the summaries will be imported into MAXQDA, where each theme will be converted into a code. This procedure will result in a list of inductively generated codes which will be combined with codes derived from our theoretical framework, thus adhering to Boyatzis' guidelines for developing a hybrid coding scheme. Phase 3 will involve the coding of the transcripts using the coding scheme developed in the previous step. Finally, in Phase 4, the patterns of relationships among the codes will be examined.

On the basis of the integrated findings and the results from these three focus groups, and using the theoretical frameworks of Causal Agency Theory and TDF, at the end of Step 1 we will be able to (a) identify the personal (i.e., behavioral) and contextual (i.e., environmental) determinants of the families' barriers to supporting self-determination, (b) refine the focus of the intervention, and (c) specify the goals of the planned intervention.

In Step 2, *formulating change objectives matrices*, for each of the intervention goals identified in Step 1, performance objectives (i.e., action statements) for expected personal and contextual outcomes will be identified and operationalized in matrices. The performance objectives will indicate what actions participants need to take (i.e., behavioral outcomes) and what contextual changes (i.e., environmental outcomes) should be made to increase support for self-determination among adolescents and young adults with intellectual disability in their home environments. In defining these performance objectives, we will take into account the findings from Phase 2 concerning potential strategies and their feasibility, as well as the feedback from the three focus groups held in Step 1. The output of Step 2 will be a series of two-dimensional matrices with the performance objectives shown in the rows, the personal and contextual determinants of the families' barriers to support self-determination in Step 1 shown in the columns, and the specific change objectives displayed within each cell. Change objectives constitute the targets of the intervention, i.e., "what needs to change in the identified determinants to achieve the specified performance objective" [71].

In Step 3, *selecting theory- and evidence-based intervention methods*, theory-based methods for inducing changes in personal and contextual determinants will be identified and translated into practical strategies that match the change objectives identified in Step 2, and these strategies are expected to help individuals achieve these objectives. The selection of the strategies will be guided by the principles of Causal Agency Theory, the family centered approach, as well as by other relevant theoretical approaches that will be identified after the development of the objectives matrices in Step 2.

Using the information from steps 1–3, in Step 4, *developing an intervention protocol*, a draft protocol for the intervention will be developed, including the definition and refinement of its components, methods, structure, and activities. These elements should contribute to achieving the change objectives developed in steps 1 and 2. The draft protocol, along with an outline of the intervention activities, will be emailed to the Phase 1 focus group participants. They will be asked to rate each of the activities in terms of relevance, effectiveness, and appropriateness, thereby helping to establish preliminary content validity for the draft intervention protocol.

The focus of Step 5, *planning the intervention implementation*, will subsequently be to develop a plan for implementing the intervention components defined in the previous step, including its feasibility. At this point, two additional focus groups of 10 individuals each with youths with intellectual disability, their families, and professionals will be held to discuss the intervention's design, development, and implementation plan generated in steps 3–4. In particular, discussions in the focus groups will focus on key implementation outcomes, how to maintain the intervention, and the extent to which the

intervention should be tailored to the particular characteristics of each adolescent and young adult. The two focus groups will follow the same data collection, transcription, and analysis procedures as the focus groups conducted in Step 1 and, ideally, they should include the same participants.

Finally, in Step 6, *planning for evaluation strategies*, a plan will be developed to evaluate the intervention's effectiveness, implementation, and acceptability. In accordance with the intervention goals defined in Step 1, the evaluation will assess the following elements: (1) potential improvements in family support mechanisms for the self-determination of adolescents and young adults with intellectual disability as a result of the intervention, as well as improvements in the self-determination outcomes of these individuals; (2) how well the intervention was implemented; and (3) how participants experienced the intervention. The AUTODDIS scale [72] will be the primary way of measuring self-determination outcomes. The use of this scale is particularly relevant to the current study because it is specifically designed to measure self-determination in people with intellectual disability, it is guided by Causal Agency Theory, and it is designed to be completed by external observers who are close to the person with intellectual disability, such as family members and professionals.

## Quality assurance

Several strategies to ensure quality will be used. First, to promote reflexivity, we will use a reflective journal in all study phases to document beliefs, values, thoughts, and feelings that may affect data collection and analysis. In addition, we will meet regularly as a research team to share decisions made, which will promote consistency and reflection. Second, to ensure accountability, we will maintain an audit trail in which we will detail the procedures followed in the development of the study so that it can be more accurately evaluated by other researchers. Audit trails have been described by Bowen [73] as an appropriate strategy for providing transparency to the decisions made in conducting a grounded theory study. Third, to increase transferability in Phase 1, we will provide a *thick description* of the findings. By providing a detailed account of the context of the findings, this strategy will allow other researchers to assess whether the study findings are transferable to other settings, thereby enhancing their generalizability [74]. Fourth, to promote credibility, the following three strategies will be used: (1) in Phase 1, investigator triangulation [75] will be achieved by involving three researchers with different disciplinary backgrounds in the grounded theory data analysis; (2) in the same phase, negative case analysis will be used by identifying "instances [in the analysis] that do not fit" [76] in order to reformulate emerging findings and add depth to the emerging theory; and (3) in all three phases, we will use peer debriefing, maintaining an exchange with external experts who will help identify potential biases and identify aspects that may have been overlooked. Fifth, we will adhere to criteria specific to the different designs employed in the three study phases. To ensure that the implementation of the grounded theory meets the standards of methodological consistency, quality, and applicability, we will use criteria specific to this research approach, as outlined by Corbin and Strauss [77]. For the participatory group concept mapping, to achieve Rosas and Kane's [78] principle of internal representational validity (i.e., "the degree to which the conceptualized model reflects the judgments made by participants in organizing information to produce the model", p. 237), we will calculate several measures of fit and similarity, including the degree of configural similarity between input and output matrices using Pearson's product-moment correlation coefficient, the stress value, and the individual sorting input. Lastly, regarding the quality of the mixed methods portion of the study, Onwuegbuzie and Johnson's [79] legitimation criteria will be used to ensure overall methodological quality, O'Cathain et al.'s [80] Good Reporting of a Mixed Methods Study (GRAMMS) guidelines will be followed to ensure overall reporting quality, and Fàbregues et al.'s [81] Mixed Methods Integration Quality Framework (MMIQF) will be used to ensure methodological and reporting quality of the integration component of the study.

## Ethics and dissemination

This study has received ethical clearance from the Institutional Review Boards of the University of Barcelona (IRB00003099) and the Universitat Oberta de Catalunya (CE25-PR32). At the beginning of each phase, participants will

sign an informed consent form. This form will inform the participants about the procedures that will be followed to preserve their anonymity and confidentiality as well as their right to ask questions about the study and to discontinue their participation at any time. For the photo elicitation procedure in Phase 1, the form will describe the use of participants' images, and they will be given the opportunity to choose which images will be used or to request that their images not be used, shared, or disseminated prior to the publication of the study. Participants will not be at any physical risk during the study. Psychologically, some of the investigators are experienced therapists who can respond to any participant's needs, concerns or sensitive issues (e.g., possible allegations of physical and mental abuse or lack of governmental support) that may arise during the research process. The methods of this study will be as non-intrusive as possible. In addition, participants will be able to withdraw from the study at any time.

To ensure anonymity, in all phases, participant names will be replaced with identification code numbers (e.g., Participant 1, Participant 2). This identification code number will also be used during data analysis. Any information that could identify participants will be deleted during this process. Only the researchers will know the identities of the participants. With respect to data management and storage, all audio and photo files will be stored on SF's computer in an encrypted, password-protected folder. In addition, a backup of the data will be stored on two encrypted external hard drives that will be accessible only to the researchers. MAXQDA files with anonymized transcriptions will be uploaded to Microsoft One-Drive by SF and shared with the rest of the research team. The interviews will be transcribed using the Happy Scribe Automated Transcription Service (https://www.happyscribe.com/). The automated transcriptions will be reviewed by SF to correct any limitations of the automated transcription (missing punctuation or incorrect words) and ensure the accuracy of the data. The Happy Scribe service includes several security measures to protect information, including network-level security monitoring and protection, DDoS protection, and data encryption. Transcriptions with all identifiers removed may be used for future projects unrelated to this study and will be made publicly available via the Internet in the Open Science Framework (OSF) data repository. This information will not include the participants' names or other personally identifiable information, ensuring that the shared data will be free of any information that could link the participants' responses to them. Audio recordings, photographs, and contact information will be deleted from SF's computer three years after the completion of the study.

The study findings will be disseminated in academic articles in peer-reviewed journals and at regional, national, and international academic and professional conferences. In addition, we will develop a report describing the study findings in plain language for professionals and families. This report will be provided directly to study participants, distributed through professional networks, and posted online for the general public.

## Study timeline

The study is expected to be completed in 2027, with the following timeline: Phase 1 (February 2025-October 2025), Phase 2 (November 2025-June 2026), Phase 3 (July 2026 – April 2027).

## Discussion

This protocol describes a mixed methods intervention study aimed at the iterative development of a family-centered intervention to support self-determination in adolescents with intellectual disability in the home environment. We anticipate certain limitations. First, due to the particularities of the sample and the potential recruitment difficulties, there is a risk of not meeting in Phase 1 the criterion of saturation in the sampling of participants. This sampling issue also applies to Phase 2, although the sampling requirements in participatory group concept mapping and intervention mapping are less strict than those in grounded theory. The literature on the first two methods does not prescribe a specific number of participants for sampling strategies. Indeed, in participatory group concept mapping, a wide range of participant numbers can be observed in published empirical studies, and having a small number of participants would not necessarily compromise the quality of the findings [60]. In contrast, grounded theory employs more strict sampling criteria, as the number and type of

participants are inherently determined by the need to further explore the unsaturated concepts within the emerging theoretical framework [77]. The problem of participant recruitment could be partially mitigated by the fact that most researchers have close contact with organizations and professionals working with people with intellectual disability, who will be crucial in identifying and recruiting participants. In addition, 20- and 10-Euro Amazon gift cards will be provided as an incentive to participants in Phases 1 and 2, respectively, which may help to increase participation. Second, and also related to the sampling strategy, although we will try to promote heterogeneity in terms of the socio-demographic characteristics of the participating families, this objective may be difficult to achieve. In addition to specific data analysis needs (i.e., saturation), the final composition of the sample will be determined both by the types of families who use the services of the professional associations to be contacted and by the willingness of those families to participate in the study.

Third, due to the specificity of the topic, it is possible that participants will not talk in detail about the self-determination processes taking place in their home environment and will instead focus their responses on other contexts, such as the school or other public spaces. With this potential problem in mind, interviewers will be proactive during the interview to ensure that information specific to the home environment is obtained. At the same time, conducting the photo elicitation interview after the first semi-structured interview might help participants emphasizing their responses on their experiences of self-determination in their home environment. Fourth, we anticipate that a possible misunderstanding of the photo elicitation instructions could lead participants to take photographs that are either irrelevant or inconsistent with the study focus. Given this possibility, we will provide participants with an instruction sheet and, in turn, send several reminder messages about both the task and the instructions during the week of photo elicitation. Fifth, the findings from Phases 1 and 2 may not be generalizable to all families with young adults and adolescents and young adults with intellectual disability. This limitation will be addressed by providing a thick description of the Phase 1 findings. Sixth, due to the length of the study, it is possible that participants from Phases 1 and 2 may not be available to participate in the intervention mapping Phase 3. In such instances, new individuals who meet our inclusion criteria and have similar characteristics to those who participated in the previous phases will be recruited. In this process, the close contact of several research team members with professional disability organizations may facilitate the recruitment of additional participants through snowball sampling techniques.

Despite these limitations, the study has several potential strengths. First, the mixed methods approach is expected to contribute to a more precise and accurate development of the intervention, as findings from earlier phases will inform phases 2 and 3 in an iterative manner. For example, using the qualitative findings from Phase 1 to generate the participatory group concept mapping statements in Phase 2 through a *building* integration strategy should allow the statements to be more closely aligned with the everyday experiences of families and also have a greater degree of "ecological" relevance in terms of the practical strategies that participants believe might need to be implemented. In addition, the integration of quantitative and qualitative findings in Phase 3 may lead to a more comprehensive needs assessment in the first step of intervention development. Second, unlike most studies in the field of intellectual disability, where the range of participants is often limited to professionals and families, Phase 2 of this study will involve people with intellectual disability as participants. We believe that listening to the voices of these people is essential in defining an intervention agenda for this type of population. Third, the study uses novel data collection methods, such as photo elicitation, to capture participants' perceptions of self-determination in the home environment in a more naturalistic and immersive way, capturing a type of information that could not be generated with only conventional interviews. This strategy is particularly important given the challenges associated with generating information that is deeply rooted in the everyday lives of households. Fourth, the study places significant emphasis on theory —specifically, Causal Agency Theory— both in its conceptualization (i.e., by considering the family and the practices within it as a modifiable context that catalyzes opportunities for self-determination of adolescents and young adults with intellectual disability) and at several of its stages to guide the development of the intervention (i.e., by incorporating components of the theory into several procedures). This use of theory is consistent with evidence suggesting that interventions based on explicit theoretical frameworks are more effective than those without [82].

In this regard, Dean et al. [31] argued that incorporating theory and specific operationalizations of self-determination into studies of families of youth with intellectual disability can increase the rigor and clarity of these studies.

To the best of our knowledge, although a few studies have documented how families naturally support the development of their sons' and daughters' self-determination in the home environment, no structured interventions have been designed specifically for this purpose. This protocol arose from the need to address this under-researched area in the field of intellectual disability, yet of great importance in the process of these individuals becoming self-determined. We believe that this study will be a valuable first step in this area of research, providing new evidence-based naturalistic knowledge about the self-determination processes that take place in the everyday inner family context of the home, and defining intervention points that will need to be implemented, tested and evaluated in a follow-up to this study.

## Acknowledgments

The authors would like to acknowledge the help of Dick Edelstein in editing the final manuscript. The authors would also like to acknowledge the use of the Paperpal tool for minor grammatical and linguistic improvements to the draft version of this manuscript.

## Author contributions

**Conceptualization:** Sergi Fàbregues, Araceli Arellano, Ahtisham Younas, Eva Vicente, Ana Berástegui, Ana Casas, Elsa Lucia Escalante-Barrios, Clara Andrés-Garriz, Cristina Mumbardó-Adam.

**Methodology:** Sergi Fàbregues, Araceli Arellano, Ahtisham Younas, Cristina Mumbardó-Adam.

**Project administration:** Sergi Fàbregues.

**Resources:** Sergi Fàbregues.

**Supervision:** Sergi Fàbregues, Araceli Arellano, Cristina Mumbardó-Adam.

**Writing – original draft:** Sergi Fàbregues, Araceli Arellano, Cristina Mumbardó-Adam.

**Writing – review & editing:** Sergi Fàbregues, Araceli Arellano, Ahtisham Younas, Eva Vicente, Ana Berástegui, Ana Casas, Elsa Lucia Escalante-Barrios, Clara Andrés-Garriz, Cristina Mumbardó-Adam.

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
