## [Decision Letter · Decision Letter 0]

8 Oct 2024

Dear Dr.  Younas,

Thank you for submitting your manuscript to PLOS ONE. After careful consideration, we feel that it has merit but does not fully meet PLOS ONE’s publication criteria as it currently stands. Therefore, we invite you to submit a revised version of the manuscript that addresses the points raised during the review process.

We look forward to receiving your revised manuscript.

Kind regards,

Hina Hadayat Ali, Ph.D

Academic Editor

PLOS ONE

Journal Requirements:

3. You indicated that you had ethical approval for your study. In your Methods section, please ensure you have also stated whether you obtained consent from parents or guardians of the minors included in the study or whether the research ethics committee or IRB specifically waived the need for their consent.

Additional Editor Comments:

The recruitment strategy for participants should be elaborated. Details on how participants were selected and the criteria for inclusion would strengthen the protocol. Consider addressing potential barriers to participation and strategies for overcoming them.

No result analyses are available to assess the technical standard. Since no data or results are presented in the manuscript, comments on the conclusions cannot be made.

Clarifying the specific outcome measures that will be used to assess self-determination would add to the protocol’s robustness.

The manuscript partially meets the ethics of experimentation and research integrity standards.

By summing up, this protocol holds significant promise for contributing to the field of intellectual disabilities. With major revision addressing recruitment strategy, result analyses, specific outcome measures, and ethics of experimentation, can effectively guide the development and implementation of the valuable intervention.

Reviewers' comments:

Reviewer's Responses to Questions

**Comments to the Author**

1. Does the manuscript provide a valid rationale for the proposed study, with clearly identified and justified research questions?

Reviewer #1: Yes

Reviewer #2: Yes

2. Is the protocol technically sound and planned in a manner that will lead to a meaningful outcome and allow testing the stated hypotheses?

Reviewer #1: Yes

Reviewer #2: Yes

3. Is the methodology feasible and described in sufficient detail to allow the work to be replicable?

Reviewer #1: Yes

Reviewer #2: Yes

4. Have the authors described where all data underlying the findings will be made available when the study is complete?

Reviewer #1: Yes

Reviewer #2: Yes

5. Is the manuscript presented in an intelligible fashion and written in standard English?

Reviewer #1: Yes

Reviewer #2: Yes

You may also provide optional suggestions and comments to authors that they might find helpful in planning their study.

Reviewer #1: Response to questions PLOS One publication criteria questions:

1. Does the study present the results of original research?

- No, this article is a concept manuscript and does not present original results.

2. Are the results reported in the article published elsewhere?

- Not applicable, as there are no results reported in the manuscript.

3. Are experiments, statistics, and other analyses performed to a high technical standard and described in sufficient detail?

- While the methodology outlined in the manuscript appears reasonable for the study, no results or analyses are available to assess the technical standard or detail of the analyses performed.

4. Are conclusions presented in an appropriate fashion and supported by the data?

- Since no data or results are presented in the manuscript, comments on the conclusions cannot be made.

5. Is the article presented in an intelligible fashion and written in standard English?

- Yes, the manuscript is presented clearly and written in standard English.

6. Does the research meet all applicable standards for the ethics of experimentation and research integrity?

- The manuscript partially meets the ethics of experimentation and research integrity standards. However, there is a potential concern regarding selection bias during recruitment of candidates for the study. Additionally, there is a possibility that conclusions drawn at the end of the study may be extrapolated from separate cohorts, which could impact the integrity of the findings.

7. Does the article adhere to appropriate reporting guidelines and community standards for data availability?

- Yes, the manuscript appears to adhere to appropriate reporting guidelines and community standards for data protection.

Reviewer Comments:

This manuscript presents a promising conceptual framework for a study that is yet to be conducted, showcasing a thoughtful approach to addressing a pertinent research problem, Self-Determination in Adolescents and Young Adults with Intellectual Disability in Home Environments.

The introduction effectively sets the stage by clearly defining the research problem and providing a comprehensive review of the relevant literature. It not only outlines existing gaps in knowledge but also convincingly justifies the significance of exploring these gaps further.

The research objectives and questions articulated in the manuscript are highly specific and closely aligned with the identified research problem. They establish a focused direction for the study, aiming to delve deep into key aspects within the scope of the research area. This clarity ensures that the study's goals are well-defined and purposeful.

In terms of theoretical framework, the manuscript proposes a solid foundation that underpins the research questions and objectives. The theoretical framework appears appropriate and relevant, offering a structured basis for understanding the phenomena under investigation. It aligns cohesively with the research goals, indicating a robust framework that could support meaningful exploration and analysis.

Methodologically, the manuscript outlines a well-considered approach, detailing aspects such as study design, sampling methods, and anticipated data collection procedures. While specific details on data analysis are not yet provided, the proposed analytical techniques suggest a thoughtful consideration of potential methodologies. The feasibility and appropriateness of the proposed methods for addressing the research questions are carefully discussed, reflecting a systematic approach to planning for future empirical investigation.

Anticipated contributions to the field are clearly articulated within the manuscript. If conducted as proposed, the study has the potential to advance knowledge through empirical investigation, inform practice in relevant contexts, and contribute to theoretical developments within the research area. The expected findings could offer valuable insights that may influence future research directions and practical applications.

While this manuscript presents a promising conceptual framework for a study yet to be conducted, several concerns warrant attention, particularly regarding potential biases in participant selection. Theoretical sampling is a reasonable approach; however, to prevent bias, it is essential to implement purposive selection, a theory-driven approach, reflexivity, triangulation, peer debriefing, transparent decision-making, and negative case analysis—all of which are crucial safeguards mentioned to varying degrees in the manuscript. These elements must be rigorously adhered to in practice to ensure methodological integrity.

A critical concern is the need for careful consideration of bias in participant selection, especially concerning socioeconomic diversity. The manuscript does not acknowledge the importance of this aspect and should explicitly outline strategies to ensure adequate representation across socioeconomic groups. Ensuring inclusivity in participant selection is crucial for minimizing biases and enhancing the study's applicability and relevance to broader populations.

Another significant concern highlighted is the potential lack of generalizability of the study findings. While grounded theory studies typically aim for theoretical rather than statistical generalizability, the manuscript should address this limitation transparently. Strategies to mitigate this concern could include detailing the contexts in which findings may be applicable or discussing the transferability of insights to similar settings.

Another crucial issue requiring attention in the manuscript is how the research team plans to address potential environmental hazards, allegations of physical and mental abuse situations or lack of support services identified during interviews and photo elicitation sessions.

In conclusion, while the manuscript shows promise with its clear conceptual framework and methodological approach, addressing these concerns will strengthen its validity and impact. Emphasizing rigorous adherence to theoretical sampling principles, ensuring socioeconomic diversity in participant selection, and transparently discussing the study's limitations are essential steps towards enhancing the manuscript's credibility and relevance in the field.

Reviewer #2: Reviewer Recommendation and Comments for Manuscript Number PONE-D-24-35031

Overall impression

This protocol successfully outlines how an intervention will be developed to support self determination in young people with an intellectual. The protocol is well-written and coherent and leaves the reader with a clear understanding of how the study will be conducted. The use of a diagram outlining the various phases contributes to this clarity. While the protocol is detailed (perhaps too detailed in parts, particularly in the early stages) the latter stages of the study are not explained to the same extent. The process for conducting Phase 3, in particular, would benefit from further detail. Subject to consideration of the issues outlined below and some reworking – specifically editing for brevity in some earlier sections and further elaboration in others - I would recommend that this manuscript progresses to the next stage.

Specific areas for improvement

1. Although a definition of intellectual disability is given in lines 195-196, the manuscript would benefit from an earlier definition bearing in mind that this is not a subject specific journal and many readers may not be familiar with intellectual disability. The term should be defined in the Introduction so that the reader understands the cohort under review from the outset.

2. Although the concept of self-determination is explained in the Introduction, an earlier placement of the definition should be considered.

3. Line 88 refers to RQ4 but there is no further reference to this research question in the manuscript. Why is this the case?

4. The Theoretical Framework section is rather long and provides much information not directly relevant to this study. It could perhaps be summarised with a focus on why the selected framework is particularly suited to this study.

5. Lines 176-177 mentions that an early literature review is recommended – has this been carried out?

6. Sampling and recruitment (line 189 on) – this section discusses participating families but it is not clear until further in the section (line 253) who the participant could be and what is meant by ‘family’. This could be explained earlier in the section to make it clearer for the reader.

7. Line 272 – data collection in Phase 1 involves a number of interviews and a photo elicitation task. It is not clear if the same person must take part in all of these or whether a different parent could participate?

8. For Phase 2, will recruitment and data collection and analysis take place concurrently as will occur for Phase 1?

9. At what point will Phase 2 recruitment conclude?

10. The paragraph from line 380 onwards is very long. The authors should consider dividing into separate paragraphs according to each step in the process.

11. Phase 3 – it is not specified how many participants will be required. Will new participants be recruited if necessary?

12. Line 453 mentions that two workshops will be held however detail around these workshops is light and not to the same level as detail provided for data collection in earlier phases of the study. For example, how many participants will be in each workshop, will the workshops be audio recorded and how will the data be analysed?

13. Similarly line 487 mentions two additional workshops but no further detail is given as to the procedure and outputs from these.

14. Line 539 mentions that participant data will be held on a computer belonging to one of the authors. This raises concerns around data protection eg implications of loss/theft of the computer? Have provisions been made to back up the data?

15. Line 571 mentions the risk of not meeting the criterion for saturation in Phase 1. What about the risk of not recruiting sufficient participants in other phases?

16. Line 590 mentions the possibility of having to recruit new participants in subsequent phases. Has the impact of including potentially different participants across phases been considered?

Minor

17. In keeping with current protocol, it would be preferable to use the full term for intellectual disability throughout the paper rather than an abbreviation

18. The format for in-text referencing (where authors are named) should be reviewed to ensure adherence to standards.

**Do you want your identity to be public for this peer review?** For information about this choice, including consent withdrawal, please see our Privacy Policy

Reviewer #1: No

Reviewer #2: No

---

## [Author Response · Author response to Decision Letter 1]

3 Dec 2024

EDITOR

1) The recruitment strategy for participants should be elaborated. Details on how participants were selected and the criteria for inclusion would strengthen the protocol. Consider addressing potential barriers to participation and strategies for overcoming them.

RESPONSE: Thank you for pointing this out. We have provided more details on the inclusion criteria for participants and on the recruitment strategies we will use in the three phases of the study. We have also highlighted the length of the study as a potential barrier to retaining participants through the phases and we mentioned the close contact between members of the research team as a resource for recruiting new participants. In addition, we discussed providing Amazon gift cards as a strategy to prevent attrition.

2) Clarifying the specific outcome measures that will be used to assess self-determination would add to the protocol’s robustness.

RESPONSE: In the final step of Phase 3 (Planning Evaluation Strategies), we made it clear that the AUTODDIS scale would be used to measure self-determination outcomes of the intervention at the time of its implementation.

3) The manuscript partially meets the ethics of experimentation and research integrity standards.

RESPONSE: In response to reviewer 1’s comment (see comment 5 below), we have clarified in the manuscript the strategies we will use to promote methodological integrity.

4) By summing up, this protocol holds significant promise for contributing to the field of intellectual disabilities. With major revision addressing recruitment strategy, result analyses, specific outcome measures, and ethics of experimentation, can effectively guide the development and implementation of the valuable intervention.

RESPONSE: Thank you for your kind comments on our manuscript.

REVIEWER 1

5) While this manuscript presents a promising conceptual framework for a study yet to be conducted, several concerns warrant attention, particularly regarding potential biases in participant selection. Theoretical sampling is a reasonable approach; however, to prevent bias, it is essential to implement purposive selection, a theory-driven approach, reflexivity, triangulation, peer debriefing, transparent decision-making, and negative case analysis—all of which are crucial safeguards mentioned to varying degrees in the manuscript. These elements must be rigorously adhered to in practice to ensure methodological integrity.

RESPONSE: Thank you for pointing this out. In the previous version of the manuscript, in the “Quality Assurance” section, we explained that some of these strategies would be used, including the use of a reflexive journal to promote reflexivity and the development of an audit trail to promote transparency. We also made clear that criteria specific to grounded theory and concept mapping would be used to ensure quality in the implementation of both approaches. In this resubmitted version, as suggested by the reviewer, we have added three additional strategies: investigator triangulation and negative case analysis during data analysis in phase 1, and peer debriefing in all three phases. In addition, the sampling strategy will be purposive in the three phases and, as explained throughout the protocol, the methodological approach is clearly guided by Causal Agency Theory.

6) A critical concern is the need for careful consideration of bias in participant selection, especially concerning socioeconomic diversity. The manuscript does not acknowledge the importance of this aspect and should explicitly outline strategies to ensure adequate representation across socioeconomic groups. Ensuring inclusivity in participant selection is crucial for minimizing biases and enhancing the study's applicability and relevance to broader populations.

RESPONSE: Thank you for mentioning this. While the aim of a grounded theory is to achieve representational and theoretical generalization rather than population generalization, we agree with the reviewer on the relevance of including a heterogeneous sample in terms of socio-demographic attributes. We have made it clear in the limitations section that we will strive to build a varied sample, but this may not be possible because participation will depend on both the types of families using the services of the associations to be contacted and the willingness of those families to participate in the study.

7) Another significant concern highlighted is the potential lack of generalizability of the study findings. While grounded theory studies typically aim for theoretical rather than statistical generalizability, the manuscript should address this limitation transparently. Strategies to mitigate this concern could include detailing the contexts in which findings may be applicable or discussing the transferability of insights to similar settings.

RESPONSE: In the “Discussion” section, we have described the limitations of the study methods for generalizing the findings. We have also explained that “thick description” (i.e., providing a detailed account of the participants’ views, meanings, and understandings) will help other researchers to assess the transferability of the study findings to other settings and contexts, thereby enhancing the generalizability of the findings.

8) Another crucial issue requiring attention in the manuscript is how the research team plans to address potential environmental hazards, allegations of physical and mental abuse situations or lack of support services identified during interviews and photo elicitation sessions.

RESPONSE: In the Ethics and Dissemination section, we have highlighted that some of the investigators are experienced therapists who can respond to any participant’s needs, concerns or sensitive issues that may arise during the research process.

REVIEWER 2

9) While the protocol is detailed (perhaps too detailed in parts, particularly in the early stages) the latter stages of the study are not explained to the same extent. The process for conducting Phase 3, in particular, would benefit from further detail.

RESPONSE: We appreciate this comment. Phase 3 will build on the findings of the previous two phases, and the final steps of this phase will also depend on the initial steps. This makes it more difficult to define concrete and definitive procedures for this phase. Despite these limitations, we have provided further details about the procedures we will follow in each of the steps. We hope that the explanation of the method is now more precise, but we are eager to provide additional information if needed.

10) Although a definition of intellectual disability is given in lines 195-196, the manuscript would benefit from an earlier definition bearing in mind that this is not a subject specific journal and many readers may not be familiar with intellectual disability. The term should be defined in the Introduction so that the reader understands the cohort under review from the outset.

RESPONSE: We have added to the beginning of the Introduction two widely known definitions of intellectual disability proposed by the American Association on Intellectual and Developmental Disabilities and the American Psychiatric Association.

11) Although the concept of self-determination is explained in the Introduction, an earlier placement of the definition should be considered.

RESPONSE: We have moved the definition of self-determination to the first paragraph, just after the definition of intellectual disability.

12) Line 88 refers to RQ4 but there is no further reference to this research question in the manuscript. Why is this the case?

RESPONSE: Thank you for bringing this to our attention. RQ4 is complementary to RQ3. We have added the mention of RQ 4 in the parts of the manuscript related to the intervention mapping method.

13) The Theoretical Framework section is rather long and provides much information not directly relevant to this study. It could perhaps be summarised with a focus on why the selected framework is particularly suited to this study.

RESPONSE: We agree with this consideration and have significantly shortened this section.

14) Lines 176-177 mentions that an early literature review is recommended – has this been carried out?

RESPONSE: We have made it clear that a nonsystematic literature review was conducted in the early stages of protocol development.

15) Sampling and recruitment (line 189 on) – this section discusses participating families but it is not clear until further in the section (line 253) who the participant could be and what is meant by ‘family’. This could be explained earlier in the section to make it clearer for the reader.

RESPONSE: Just after mentioning that families will be sampled in this study, we have clarified that participants in each family will include one or both parents or the primary caregiver.

16) Line 272 – data collection in Phase 1 involves a number of interviews and a photo elicitation task. It is not clear if the same person must take part in all of these or whether a different parent could participate?

RESPONSE: While the number of participants from each family may vary between interviews (e.g., the second interview may include both the mother and father, while the third interview may include only the mother), at least one participant (i.e., the same person) must participate in all three interviews. We have made this clear in the manuscript.

17) For Phase 2, will recruitment and data collection and analysis take place concurrently as will occur for Phase 1?

RESPONSE: In Phase 2, recruitment will occur prior to the implementation of the first step of the method. As reported in the concept mapping literature, it is likely that this number will decrease in subsequent steps. However, we will strive to have no less than 30-35 participants in these steps (a number consistent with recommendations in the literature), recruiting new participants if necessary. Data analysis will be conducted in step 4 after data collection (steps 1, 2 and 3) has been completed. We have made this clear in the manuscript.

18) At what point will Phase 2 recruitment conclude?

RESPONSE: See response to comment number 17.

19) The paragraph from line 380 onwards is very long. The authors should consider dividing into separate paragraphs according to each step in the process.

RESPONSE: We have shortened the paragraph by splitting it into two paragraphs that refer to step 1 (first paragraph) and to steps 2 and 3 (second paragraph).

20) Phase 3 – it is not specified how many participants will be required. Will new participants be recruited if necessary?

RESPONSE: We have explained that three focus groups of 10 participants each will be held in step 1 and two with ideally the same participants in step 2. Therefore, the total number of participants should be around 30, subject to increase if participants from the first focus group are unavailable for the second. If necessary, new participants with similar characteristics to those in the previous phases will be recruited.

21) Line 453 mentions that two workshops will be held however detail around these workshops is light and not to the same level as detail provided for data collection in earlier phases of the study. For example, how many participants will be in each workshop, will the workshops be audio recorded and how will the data be analysed?

RESPONSE: We agree with this comment. To be more methodologically precise, we have replaced the term “workshop” with “focus groups”. We have also provided more details about the total number of focus group participants (n=30) and the procedures we will use to conduct, transcribe, and analyze the focus group data.

22) Similarly line 487 mentions two additional workshops but no further detail is given as to the procedure and outputs from these.

RESPONSE: We have made it clear that the same sampling, data collection, and analysis procedures from the focus groups conducted in Step 1 will be followed in these additional two focus groups conducted in Step 5. We have also provided more details about the type of information that will be collected in these two focus groups.

23) Line 539 mentions that participant data will be held on a computer belonging to one of the authors. This raises concerns around data protection eg implications of loss/theft of the computer? Have provisions been made to back up the data?

RESPONSE: Thank you for this suggestion. We have added that a backup of the data will be stored on two encrypted external drives that will be accessible only to the researchers.

24) Line 571 mentions the risk of not meeting the criterion for saturation in Phase 1. What about the risk of not recruiting sufficient participants in other phases?

RESPONSE: We explicitly stated that this risk applies to the other phases as well. However, we also mentioned that grounded theory is characterized by stricter sampling criteria than participatory group concept mapping and intervention mapping.

25) Line 590 mentions the possibility of having to recruit new participants in subsequent phases. Has the impact of including potentially different participants across phases been considered?

RESPONSE: In Phase 2, this should not present a significant challenge because the sample size needs to be larger than in Phase 1, thereby requiring the recruitment of new participants. With respect to Phase 3, if participants from Phases 1 and 2 do not agree to participate or are unavailable, we will recruit new individuals who meet our inclusion criteria and have similar characteristics to those in the previous phases. We have made this clear in the manuscript text.

26) In keeping with current protocol, it would be preferable to use the full term for intellectual disability throughout the paper rather than an abbreviation

RESPONSE: We have replaced abbreviations of terms throughout the manuscript, including those related to intellectual disability, research questions, grounded theory, participatory group concept mapping, and intervention mapping, with the full term.

27) The format for in-text referencing (where authors are named) should be reviewed to ensure adherence to standards.

RESPONSE: In-text referencing has been revised to ensure compliance with the Verdana referencing style. We used EndNote to insert the references into the manuscript.

---

## [Decision Letter · Decision Letter 1]

2 Jan 2025

Dear Dr.  Ahtisham Younas,

Thank you for submitting your manuscript to PLOS ONE. After careful consideration, we feel that it has merit but does not fully meet PLOS ONE’s publication criteria as it currently stands. Therefore, we invite you to submit a revised version of the manuscript that addresses the points raised during the review process.

There is concern about the lack of specific strategies to ensure representation across socioeconomic groups, even though the authors aim for a varied sample. The limitation of the study's findings being applicable only to participants similar to those in the study is viewed as a significant drawback to an otherwise robust design.

We look forward to receiving your revised manuscript.

Kind regards,

Hina Hadayat Ali, Ph.D

Academic Editor

PLOS ONE

Journal Requirements:

Reviewers' comments:

Reviewer's Responses to Questions

**Comments to the Author**

1. Does the manuscript provide a valid rationale for the proposed study, with clearly identified and justified research questions?

Reviewer #1: Yes

Reviewer #2: Yes

2. Is the protocol technically sound and planned in a manner that will lead to a meaningful outcome and allow testing the stated hypotheses?

Reviewer #1: Yes

Reviewer #2: Yes

3. Is the methodology feasible and described in sufficient detail to allow the work to be replicable?

Reviewer #1: Yes

Reviewer #2: Yes

4. Have the authors described where all data underlying the findings will be made available when the study is complete?

Reviewer #1: Yes

Reviewer #2: Yes

5. Is the manuscript presented in an intelligible fashion and written in standard English?

Reviewer #1: Yes

Reviewer #2: Yes

You may also provide optional suggestions and comments to authors that they might find helpful in planning their study.

Reviewer #1: Thank you for the opportunity to review the revised manuscript entitled "Development of Family-Centered Interventions to Support Self-Determination in Adolescents and Young Adults with Intellectual Disability in Home Environments: Protocol for Multistage Mixed Methods Design." After carefully reviewing the updated version, I am pleased to report that the authors have diligently addressed all of my previous recommendations, significantly enhancing the clarity, methodological rigor, and relevance of the work.

The authors have taken considerable steps to incorporate essential methodological safeguards to ensure the integrity and reliability of the study. The inclusion of strategies such as investigator triangulation, purposive sampling across all phases, peer debriefing, and negative case analysis demonstrates their commitment to addressing potential biases in participant selection and data interpretation. Additionally, the use of a reflexive journal and audit trail, alongside criteria aligned with grounded theory and concept mapping, reflects a rigorous and transparent approach to quality assurance. These enhancements meaningfully address the critical concerns outlined in earlier feedback and strengthen the methodological framework of the study.

Furthermore, the authors have thoughtfully expanded on the importance of ensuring socioeconomic diversity within the sample, acknowledging potential limitations while striving for a heterogeneous participant pool. Their use of "thick description" to detail participants' views, meanings, and understandings enhances the potential transferability of findings to similar settings. They have also explicitly discussed the ethical considerations of addressing environmental hazards, allegations of abuse, and lack of support services, emphasizing the expertise of their research team in managing sensitive issues that may arise during the research process.

In light of these comprehensive revisions, I am confident in recommending this manuscript for publication. It represents an important contribution to the field, offering a robust, family-centered framework to support self-determination in adolescents and young adults with intellectual disabilities. The study not only addresses a critical gap in the literature but also exemplifies methodological rigor and ethical responsibility.

Sincerely,

José A. Acosta MD, MBA, MPH

Reviewer #2: This manuscript represents a detailed and well-written protocol for an original study. That people with an intellectual disability will be involved as active participants is particularly welcome.

While most of the earlier review comments have been addressed, I feel that concerns remain about the potential for bias in participant selection. Though the authors have responded that they will strive to build a varied sample, there are no specific strategies outlined which will seek to ensure adequate representation across socioeconomic groups.

I feel that the acknowledged limitation that the 'aim of this study is to develop an intervention that can be used by individuals similar to those who participated in our study, rather than to generalize the findings beyond the group of participants' represents a serious limitation to an otherwise robust design.

Could greater efforts not be made to target particular groups and these strategies described? For example could the institutions and special education schools contacted not be drawn from a cross-section of socioeconomic settings to ensure there is some degree of representativeness?

**Do you want your identity to be public for this peer review?** For information about this choice, including consent withdrawal, please see our Privacy Policy

Reviewer #1: No

Reviewer #2: No

---

## [Author Response · Author response to Decision Letter 2]

20 Feb 2025

In the Sampling and Recruitment subsection, we explained that we will aim to achieve socioeconomic diversity using the stratified random sampling strategy as suggested by Patton (2014). In line with this strategy, when identifying families to be recruited, we will select institutions and special education schools from different socioeconomic areas using socioeconomic indices in each region, if available. We agree with the reviewer that such a strategy will help to broaden the scope of the intervention to be developed in phase 3 of the study, particularly in terms of the population to which the intervention is applicable. Accordingly, we have removed the limitation that the results will be only applicable to a specific profile of individuals.

---

## [Decision Letter · Decision Letter 2]

12 Mar 2025

Dear Dr. Younas,

Thank you for submitting your manuscript to PLOS ONE. After careful consideration, we feel that it has merit but does not fully meet PLOS ONE’s publication criteria as it currently stands. Therefore, we invite you to submit a revised version of the manuscript that addresses the points raised during the review process.

Minor corrections are required to ensure consistency in the discussion of family support for self-determination (Lines 57–60) and to align the study timeline with publication requirements (Line 671). The authors should clarify contradictory statements and exclude pilot study data from the results.

We look forward to receiving your revised manuscript.

Kind regards,

Hina Hadayat Ali, Ph.D

Academic Editor

PLOS ONE

Journal Requirements:

Reviewers' comments:

Reviewer's Responses to Questions

**Comments to the Author**

1. Does the manuscript provide a valid rationale for the proposed study, with clearly identified and justified research questions?

Reviewer #1: Yes

Reviewer #2: Yes

2. Is the protocol technically sound and planned in a manner that will lead to a meaningful outcome and allow testing the stated hypotheses?

Reviewer #1: Yes

Reviewer #2: Yes

3. Is the methodology feasible and described in sufficient detail to allow the work to be replicable?

Reviewer #1: Yes

Reviewer #2: Yes

4. Have the authors described where all data underlying the findings will be made available when the study is complete?

Reviewer #1: Yes

Reviewer #2: Yes

5. Is the manuscript presented in an intelligible fashion and written in standard English?

Reviewer #1: Yes

Reviewer #2: Yes

You may also provide optional suggestions and comments to authors that they might find helpful in planning their study.

Reviewer #1: Overall, this manuscript has improved significantly since the first time I reviewed it. There are only a couple of issues that need to be addressed at this point which are:

Lines 57 to 60: The authors should provide some examples from their literature review on how families promote self-determination within the home. The statement " While most of the studies reviewed focused on the family’s view of the importance of self-determination to their children and to adolescents with disabilities of this sort, only a few of these provided evidence on how families promoted self-determination within the home." contradicts the statement in the discussion section starting in line 757 which states “To the best of our knowledge, no interventions have addressed the issue of supporting families of adolescents and young adults with intellectual disability to develop their self-determination in the home environment.” The authors should make sure that these statements are congruent.

Line 671: The study timeline should be updated since consideration for publication requires that study protocols be approved prior to the collection of any data. Data from the Pilot study should not be included as part of the results of this protocol.

Thank you for giving me the opportunity to review this manuscript.

Reviewer #2: The last remaining issue has been addressed satisfactorily. Therefore I recommend this manuscript for publication. Congratulations to the authors.

**Do you want your identity to be public for this peer review?** For information about this choice, including consent withdrawal, please see our Privacy Policy

Reviewer #1: No

Reviewer #2: No

---

## [Author Response · Author response to Decision Letter 3]

19 Apr 2025

EDITOR

(1) Minor corrections are required to ensure consistency in the discussion of family support for self-determination (Lines 57–60) and to align the study timeline with publication requirements (Line 671). The authors should clarify contradictory statements and exclude pilot study data from the results.

Response:

Thank you for your comments. We have addressed these issues in our responses to reviewer 1. Please see our responses below.

REVIEWER 1

(1) Overall, this manuscript has improved significantly since the first time I reviewed it. There are only a couple of issues that need to be addressed at this point which are:

Response:

Thank you for your kind words.

(2) Lines 57 to 60: The authors should provide some examples from their literature review on how families promote self-determination within the home. The statement " While most of the studies reviewed focused on the family’s view of the importance of self-determination to their children and to adolescents with disabilities of this sort, only a few of these provided evidence on how families promoted self-determination within the home." contradicts the statement in the discussion section starting in line 757 which states “To the best of our knowledge, no interventions have addressed the issue of supporting families of adolescents and young adults with intellectual disability to develop their self-determination in the home environment.” The authors should make sure that these statements are congruent.

Response:

Thank you for your comments. In lines 73-80, we have cited the work of Brotherson et al. (2008) and Erwin et al. (2009), which identified a number of strategies to promote self-determination in the home environment. Some of these strategies are mentioned in the manuscript.

To make the two statements more congruent, in lines 774-777, we have clarified in the second statement that while a few studies have been conducted on the topic of self-determination of people with ID in the home context, no interventions have been designed for this purpose. In other words, the studies that have been conducted have been merely exploratory, and their findings have not led to the development of an intervention.

(3) Line 671: The study timeline should be updated since consideration for publication requires that study protocols be approved prior to the collection of any data. Data from the Pilot study should not be included as part of the results of this protocol.

Response:

In lines 683-685, we have updated the study timeline by indicating the date when we started the data collection (February 2005) and adjusting the rest of the dates. Thank you for your suggestion. However, we would like to point out that the criteria for publication of protocols in Plos One is that the manuscript must “be submitted before recruitment of participants or collection of data for the study is complete” (https://journals.plos.org/plosone/s/submission-guidelines#loc-study-protocols). Therefore, not having started the data collection is not a requirement for publishing protocols in the journal.

Thank you for your comment regarding our reference to the pilot results. We have removed this information from the manuscript.

REVIEWER 2

(1) The last remaining issue has been addressed satisfactorily. Therefore I recommend this manuscript for publication. Congratulations to the authors.

Response:

Thank you for your kind comment.

---

## [Decision Letter · Decision Letter 3]

21 May 2025

Development of a family-centered intervention to support self-determination in adolescents and young adults with intellectual disability in home environments: Protocol for a multistage mixed methods design

PONE-D-24-35031R3

Dear Ahtisham Younas, Ph.D,

We’re pleased to inform you that your manuscript has been judged scientifically suitable for publication and will be formally accepted for publication once it meets all outstanding technical requirements.

Kind regards,

Hina Hadayat Ali, Ph.D

Academic Editor

PLOS ONE

Additional Editor Comments (optional):

Reviewers' comments:

Reviewer's Responses to Questions

**Comments to the Author**

1. Does the manuscript provide a valid rationale for the proposed study, with clearly identified and justified research questions?

Reviewer #1: Yes

Reviewer #2: Yes

2. Is the protocol technically sound and planned in a manner that will lead to a meaningful outcome and allow testing the stated hypotheses?

Reviewer #1: Yes

Reviewer #2: Yes

3. Is the methodology feasible and described in sufficient detail to allow the work to be replicable?

Reviewer #1: Yes

Reviewer #2: Yes

4. Have the authors described where all data underlying the findings will be made available when the study is complete?

Reviewer #1: Yes

Reviewer #2: Yes

5. Is the manuscript presented in an intelligible fashion and written in standard English?

Reviewer #1: Yes

Reviewer #2: Yes

You may also provide optional suggestions and comments to authors that they might find helpful in planning their study.

Reviewer #1: Your manuscript revisions have thoroughly addressed all my concerns. I look forward to seeing the results of this very important study.

Reviewer #2: All comments have been submitted previously. I have no further comments. Well done to the authors for this interesting study.

**Do you want your identity to be public for this peer review?** For information about this choice, including consent withdrawal, please see our Privacy Policy

Reviewer #1: **Yes: ** José A. Acosta MD, MBA, MPH

Reviewer #2: No

---

## [Editor Report · Acceptance letter]

PONE-D-24-35031R3

PLOS ONE

Dear Dr. Younas,

I'm pleased to inform you that your manuscript has been deemed suitable for publication in PLOS ONE. Congratulations! Your manuscript is now being handed over to our production team.

Kind regards,

on behalf of

Dr. Hina Hadayat Ali

Academic Editor

PLOS ONE